# Percutaneous Computed Tomography-Guided Oxygen-Ozone (O_2_O_3_) Injection Therapy in Patients with Lower Back Pain—An Interventional Two-Year Follow-Up Study of 321 Patients

**DOI:** 10.3390/diagnostics13213370

**Published:** 2023-11-02

**Authors:** Kristina Davidovic, Sebastian Cotofana, Stephan Heisinger, Slavica Savic, Michael Alfertshofer, Tatjana Antonić, Sanja Jovanović, Marko Ercegovac, Mario Muto, Danilo Jeremić, Aleksandar Janićijević, Lukas Rasulić, Vesna Janošević, Lidija Šarić, Deborah Chua, Dragan Masulovic, Ružica Maksimović

**Affiliations:** 1Center for Radiology and Magnetic Resonance Imaging, University Clinical Center of Serbia, 11000 Belgrade, Serbia; dr.kristina.davidovic@gmail.com (K.D.);; 2Department of Dermatology, Erasmus Medical Centre, 3015 GD Rotterdam, The Netherlands; 3Centre for Cutaneous Research, Blizard Institute, Queen Mary University of London, London E1 4NS, UK; 4Department of Plastic and Reconstructive Surgery, Guangdong Second Provincial General Hospital, Guangzhou 510317, China; 5Department of Orthopedics and Trauma Surgery, Medical University of Vienna, 1090 Vienna, Austria; 6Medical Faculty, University of Belgrade, 11120 Belgrade, Serbia; 7Division of Hand, Plastic and Aesthetic Surgery, LMU University Hospital, 80336 Munich, Germany; 8Acibadem Bel Medic, 11000 Belgrade, Serbia; 9Clinic of Neurology, University Clinical Centre of Serbia, 11000 Belgrade, Serbia; 10Neuroradiology Department, Ospedale Cardarelli Napoli, 80131 Napoli, Italy; 11Institute for Orthopedic Surgery “Banjica”, 11000 Belgrade, Serbia; 12Clinic of Neurosurgery, University Clinical Centre of Serbia, 11000 Belgrade, Serbia; 13MH Plastic Surgery, Singapore 238859, Singapore

**Keywords:** low back pain, ozone, corticosteroids, magnet resonance imaging, computed tomographic imaging, lumbosacral spine, oxygen-ozone therapy, minimally invasive treatment, spine degeneration

## Abstract

Objectives: To assess the effect of oxygen-ozone therapy guided by percutaneous Computed Tomography (CT) compared to corticosteroids in individuals experiencing lower back pain (LBP) not attributed to underlying bone-related issues. Methods: A total of 321 patients (192 males and 129 females, mean age: 51.5 ± 15.1 years) with LBP were assigned to three treatment groups: group A) oxygen-ozone only, group B) corticosteroids only, group C) oxygen-ozone and corticosteroids. Treatment was administered via CT-guided injections to the intervertebral disc (i.e., intradiscal location). Clinical improvement of pain and functionality was assessed via self-reported pain scales and magnetic resonance (MR) and CT imaging. Results: At all follow-up times, the mean score of the numeric rating scale and the total global pain scale (GPS) of study groups receiving oxygen-ozone (groups A and C) were statistically significantly lower than the study group receiving corticosteroids only (group B), with *p* < 0.001. There was a statistically significant difference between groups A and C at 30 days for the numeric rating scale. Conclusions: The percutaneous application of oxygen-ozone in patients with LBP due to degeneration of the lumbosacral spine showed long-lasting significant pain reduction of up to two years post-treatment when compared to corticosteroids alone. Combination therapy of oxygen-ozone and corticosteroids can be useful as corticosteroids showed statistically significant improvement in LBP earlier than the oxygen-ozone-only treatment.

## 1. Introduction

In the year 2019, 7.6% of the global population was estimated to suffer from lower back pain (LBP). Between 1990 and 2019, the years of healthy life lost due to disability (YLD) increased tremendously by 49.9%, from 42.5 million YLDs in 1990 to 63.7 million YLDs in 2019. Of all conditions studied in the Global Burden of Disease, LBP was shown to be the leading cause of YLDs [1,2,3,4,5,6,7].

While LBP can present without any pathological correlate and may resolve spontaneously, more severe cases often result from underlying degenerative processes of the lumbosacral spine. Loss of structural integrity and function of the spine can be secondary to a myriad of factors: age-related osteoporotic bony depletion, pathological load distribution (e.g., in scoliosis), tumors, infections, arthritis, and intervertebral disc degeneration [8,9,10,11,12]. Multifactorial causes can ultimately lead to disc herniation with increased pressure, thereby causing mechanical stress on adjacent spinal nerve roots. 

Analogous to the classic egg-and-hen dilemma, the degenerative processes of the spine and lower back pain are closely interconnected, with each potentially causing and exacerbating the other.

Depending on the clinical presentation and diagnostic findings, therapeutic options for LBP may include conservative, open surgical, and percutaneous treatments [13,14,15,16,17]. Conservative treatment options (e.g., rest, medication, physical therapy) are chosen for mild to moderate cases in which spontaneous remission of pain is likely. While open surgical therapy often yields satisfactory outcomes, it does come with inherent risks, particularly when dealing with elderly patients [18,19,20]. Postoperative complications such as “Failed Back Surgery Syndrome” have been reported to range between 10 and 40% in surgical treatments and have, therefore, led to increasing interest in alternatives, including percutaneous procedures [21,22,23,24].

Percutaneous, minimally invasive treatment options include mechanical, thermal, and chemical decompression, as well as biomaterial implantation, cellular therapies, and the administration of oxygen-ozone via CT-guided injections. The latter has been shown to significantly reduce pain and increase function with low rates of complications while being affordable [25,26,27]. Anatomic sites for percutaneous minimally invasive oxygen-ozone injections are the intervertebral disc or the structures located within the intervertebral foramen via an interdiscal, translaminar, transforaminal, paraforaminal, or posterolateral approach. The use of oxygen-ozone injections demonstrates emerging popularity as a form of minimally invasive treatment for LBP, either as part of conservative treatment options before surgery or when surgery is contra-indicated [28].

Previous research has suggested that both epidural corticosteroid administration and intradiscal oxygen-ozone infiltration can yield favorable clinical outcomes in both the short and long term. Nevertheless, each approach has its own set of advantages and limitations with respect to factors such as treatment onset, duration of relief, and treatment effectiveness [17,29,30]. Although corticosteroids have traditionally shown faster initial results, oxygen-ozone therapy is believed to offer more extended benefits in the long term. Yet, thus far, no study has examined the effectiveness of these two modalities, either individually or in combination, through a comprehensive, interventional, imaging-based longitudinal study design.

Therefore, the objective of this study is to investigate the effect of percutaneous CT-guided injections with oxygen-ozone (single therapeutic arm), cortico-steroid (single therapeutic arm), and their combination (dual therapeutic arm) for LBP in a comparative longitudinal (two-year follow-up) study design focussing on pain reduction and functionality in patients with degenerative changes of the lumbosacral spine.

## 2. Material and Methods

### 2.1. Study Design

This study was designed as an interventional prospective randomized study that investigated the difference in the long-term (2 years) outcome of three different treatment modalities: oxygen-ozone (single therapeutic arm), corticosteroid (single therapeutic arm), and their combination (dual therapeutic arm) for LBP. Study participants were informed about the aims and the methodology of this study. Written informed consent for the use of their personal and clinical data for research purposes was obtained prior to their study inclusion.

The study was conducted between January 2019 and June 2022 and received institutional review board approval from the Emergency Room Department, University Clinical Centre of Serbia, under the approval number 1R23K82MS1224/2018.

### 2.2. Study Sample and Inclusion/Exclusion Criteria

The study sample comprised a total of *n* = 321 patients (192 males and 129 females) with a mean age of 51.5 ± 15.1 years (range: 18–91 years). Patients included in this study were consecutive patients of the Department of Emergency Radiology, Center for Radiology and Magnetic Resonance Imaging, Clinical Center of Serbia, Belgrade, Serbia.

The inclusion criteria were as follows: Age ≥ 18 years old; clinical presentation of neurological symptoms (including dermatomal paresthesia or radiating pain such as sciatica); pain in the lumbosacral region most likely, but not necessarily associated with discal pathologies and LBP refractory to oral analgesia and physical therapy. 

Exclusion criteria comprised spinal motor deficits, bleeding disorders, favism, diabetic neuropathy, cauda equina syndrome, pregnancy, autoimmune diseases, and/or allergies to the treatment material. Additional exclusion criteria are pathologic bony changes like spinal canal stenosis, osteophytic compression of the spine or spinal nerve, lumbar disc sequestration, or any other pathological changes requiring surgical attention. (Figure 1) Previous medical history of patients is summarized in Table 1. 

### 2.3. Treatment Protocol

To evaluate treatment outcomes, patients were assigned to one of three treatment groups of CT-guided lumbar transforaminal injections: oxygen-ozone (single therapeutic arm), cortico-steroid (single therapeutic arm), and their combination (dual therapeutic arm) for LBP. Patients who were treated with oxygen-ozone only (Group A) and corticosteroids only (Group B) were assigned following a matching process for patients’ demographic data and diagnosis. 

-Group A received percutaneous injections of 5–10 mL of an oxygen-ozone gas mixture (2% ozone in a concentration of 30–40 mg/mL and 98% oxygen);-Group B received percutaneous injections of corticosteroids (1 mL of long-acting corticosteroids and 1 mL of lidocaine chloride);-Group C received combined treatment with percutaneous injections of both oxygen-ozone gas mixture and corticosteroids (2% ozone in a concentration of 30–40 mg/mL and 98% oxygen + 1 mL of long-acting corticosteroids and 1 mL of lidocaine chloride).

Patients were treated based on a combination of their individual preferences and the medical indications deemed suitable by the treating physician, ensuring an individualized and patient-centric approach to care while aligning with established therapeutic guidelines.

The oxygen-ozone gas mixture was produced using an ozone generator, OZO2 (Alnitec S.R.L., Cremosano, Italy), and the concentration of the oxygen-ozone gas mixture was monitored throughout the process. The corticosteroids used in this study were Dexason 4 mg/mL (Galenika, Belgrade, Serbia). Corticosteroid injections as a comparator were chosen due to the similar level of invasiveness and their widespread use in the treatment of LBP.

Patients did not receive any pre-procedural or post-procedural medications. The patients were discharged within four hours after the procedure. Following the procedure, patients were advised to refrain from engaging in strenuous physical activities for a period of 1–3 months and to avoid undergoing physical therapy.

The injections were performed under sterile conditions. The lumbar region was disinfected with chlorohexidine, and sterile drapes were placed. The patient was positioned in a prone position on the CT sliding table (Aquilion PRIME 64, Toshiba Medical Systems Corporation, Tochigi, Japan). CT sections were obtained using the following parameters: voltage 120 kV, tube current 250 mA, and slice thickness 0.5 mm. 

Depending on the respective patient’s body habitus, a 15–20 cm long 18–22 gauge needle (Sterican^®^ Safety, B Braun, Melsungen, Germany) was used. Under CT guidance, the injection was performed by experienced interventional radiologists applying the techniques described by Wagner [31]. The injection procedure was limited to a maximum of 15 s due to the inherent instability of oxygen-ozone, which begins to degrade after approximately 20 s [32]. A post-procedural CT scan (Revolution^TM^ Discovery^TM^ CT, General Electric) identifies and confirms the accurate distribution of gas (Figure 2, Figure 3 and Figure 4).

### 2.4. Self-Reported Pain Scales

At baseline, patients were clinically examined, and pain perception of LBP was rated using the self-reported numeric rating scale (NRS) and the global pain scale (GPS). Follow-up periods were 30 days, 90 days, 180 days, 1 year, 1.5 years, and 2 years after the treatment.

The improvement in pain control after therapy was assessed using the self-reported NRS and the GPS. The NRS is an 11-point Likert scale ranging from 0 = “No pain” to 10 = “Debilitating pain”. The GPS allows one to analyze the perception of pain in a holistic way and enables one to evaluate different aspects of pain since the total GPS score is comprised of four different categories: “Your Pain” (i.e., physical pain), “Your Feelings” (i.e., severity of negative feelings), “Your Clinical Outcomes” (i.e., ability to find rest), and “Your Activities” (i.e., ability to perform daily life activities). Each category consists of five questions, which can be rated on an 11-point Likert scale ranging from 0 = “No pain” to 10 = “Extreme pain”. The total GPS score is calculated by adding all points and dividing them by two.

### 2.5. Statistical Analysis

Primary outcome parameters were average pain levels as assessed by the NRS and by the GPS. Results were compared from baseline and between the three treatment groups. Parameters were tested for normal distribution using the Shapiro-Wilk test. Due to non-normal data distribution, non-parametric tests were used for statistical analysis. Differences between the three study groups were calculated using the Kruskal-Wallis test, while differences between two study groups were calculated using Mann-Whitney U test. For better readability of data, data of self-reported pain scales is given as mean value and standard deviation. The statistical analysis was run using SPSS Statistics 25 (IBM), and differences were considered statistically significant at a two-tailed *p*-value of ≤0.05.

## 3. Results

### 3.1. Procedural Technical Outcome

The performed injection procedure was successful in 100% of the cases as defined by product distribution into the intervertebral disc (i.e., intradiscal location) located within the intervertebral space as visualized during CT scanning.

### 3.2. Patient Demographic Data

The study sample consisted of a total of *n* = 321 patients (192 males, 129 females) with a mean age of 51.5 ± 15.1 years (range: 18–91 years). Lower back pain for less than 6 months was reported by 17.1% of all patients (*n* = 55), 6–12 months by 14.6% (*n* = 47), 12–24 months by 16.5% (*n* = 53), while the majority reported to have had LBP for more than 24 months with 51.7% (*n* = 166) of all patients included in this study, regardless of treatment group. Regarding patient history, 48.9% (*n* = 157) had previous therapy, 7.5% (*n* = 24) had previous surgery, and 51.4% (*n* = 165) had physical therapy for their symptoms. No complications were reported during the follow-up period of this study (Table 1).

### 3.3. Numeric Rating Scale

At baseline, the mean NRS score was 7.14 ± 2.3 for the oxygen-ozone-only group, 6.94 ± 2.4 for the corticosteroids-only group, and 6.58 ± 2.2 for the oxygen-ozone + corticosteroids group, with *p* = 0.229. After 30 days, the mean NRS score was 2.22 ± 2.3 for the oxygen-ozone-only group, 5.00 ± 2.2 for the corticosteroids-only group, and 1.50 ± 2.0 for the oxygen-ozone + corticosteroids group, with *p* < 0.001. After 90 days, the mean NRS score was 1.28 ± 2.1 for the oxygen-ozone-only group, 5.80 ± 1.9 for the corticosteroids-only group, and 1.08 ± 1.7 for the oxygen-ozone + corticosteroids group, with *p* < 0.001. After 180 days, the mean NRS score was 1.53 ± 2.3 for the oxygen-ozone-only group, 6.00 ± 2.0 for the corticosteroids-only group, and 1.18 ± 2.0 for the oxygen-ozone + corticosteroids group, with *p* < 0.001. After one year, the mean NRS score was 1.11 ± 2.3 for the oxygen-ozone-only group, 6.09 ± 2.0 for the corticosteroids-only group, and 1.15 ± 2.0 for the oxygen-ozone + corticosteroids group, with *p* < 0.001. After one and a half years, the mean NRS score was 1.08 ± 2.2 for the oxygen-ozone-only group, 6.11 ± 1.9 for the corticosteroids-only group, and 1.10 ± 2.0 for the oxygen-ozone + corticosteroids group, with *p* < 0.001. At the final follow-up after two years, the mean NRS score was 1.03 ± 2.2 for the oxygen-ozone-only group, 6.26 ± 1.8 for the corticosteroids-only group, and 1.09 ± 2.0 for the oxygen-ozone + corticosteroids group, with *p* < 0.001 (Figure 5).

The only statistically significant difference between oxygen-ozone only (group A) and oxygen-ozone + corticosteroids study group (group C) was found for the mean NRS score 30 days after the treatment, with *p* = 0.047 as group C showed a greater reduction in the NRS score.

### 3.4. Global Pain Scale

At baseline, the mean total GPS score was 61.1 ± 22.7 for the oxygen-ozone-only group, 57.7 ± 21.2 for the corticosteroids-only group, and 57.9 ± 22.2 for the oxygen-ozone + corticosteroids group, with *p* = 0.604. After 30 days, the mean score on the pain scale was 11.3 ± 11.3 for the oxygen-ozone-only group, 42.3 ± 22.4 for the corticosteroids-only group, and 9.5 ± 14.5 for the oxygen-ozone + corticosteroids group, with *p* < 0.001. After 90 days, the mean score on the pain scale was 7.8 ± 10.4 for the oxygen-ozone-only group, 51.1 ± 16.6 for the corticosteroids-only group, and 7.0 ± 12.1 for the oxygen-ozone + corticosteroids group, with *p* < 0.001. After 180 days, the mean score on the pain scale was 9.3 ± 12.3 for the oxygen-ozone-only group, 52.9 ± 19.1 for the corticosteroids-only group, and 6.8 ± 13.6 for the oxygen-ozone + corticosteroids group, with *p* < 0.001. After one year, the mean score on the pain scale was 3.7 ± 6.9 for the oxygen-ozone-only group, 54.5 ± 19.3 for the corticosteroids-only group, and 4.4 ± 8.8 for the oxygen-ozone + corticosteroids group, with *p* < 0.001. After one and a half years, the mean score on the pain scale was 3.2 ± 6.8 for the oxygen-ozone-only group, 55.0 ± 18.9 for the corticosteroids-only group, and 3.8 ± 7.6 for the oxygen-ozone + corticosteroids group, with *p* < 0.001. After two years, the mean score on the pain scale was 2.0 ± 4.4 for the oxygen-ozone-only group, 62.5 ± 17.0 for the corticosteroids-only group, and 3.3 ± 6.5 for the oxygen-ozone + corticosteroids group, with *p* < 0.001. (Figure 6) The scores for the four different categories (“Your Pain”, “Your Feelings”, “Your Clinical Outcomes”, and “Your Activities”) of the Global Pain Scale can be seen in Table 2 (Figure 7).

No adverse events were observed during the treatment procedure or the two-year follow-up period.

## 4. Discussion

This interventional prospective study sought to explore the effects of percutaneous oxygen-ozone treatment on clinical outcomes in patients with LBP when compared to corticosteroids. Since first being described as a paravertebral injection in 1989 by Verga [33] and later as an intradiscal injection in 1998 by Muto et al. [34], the use of oxygen-ozone for the treatment of (sub-)acute and chronic LBP has been analyzed in several prospective and retrospective studies [25,29,30,32,35,36,37,38,39,40,41]. However, there is a paucity of data available on the long-term effects of oxygen-ozone percutaneous injection therapy, especially in combination with other treatment strategies such as corticosteroids in a single-treatment or dual-treatment randomized observational clinical study design with two-year follow-up [40].

The results of this study confirm the consensus established by previous research that percutaneous oxygen-ozone application can lead to clinical improvement, particularly in reducing LBP and enhancing functionality. In our study, group A and group C, which both received oxygen-ozone treatment, demonstrated a statistically significant decrease in the NRS and the GPS when compared to group B, which has been treated with corticosteroids alone. Notably, no significant change was observed in group B over the two-year follow-up period when compared to baseline scores, with a mean of 6.9 at baseline and 6.3 after the two-year follow-up period.

Interesting observations were made regarding the onset and dynamics of the treatment agents investigated in this study: Statistically significant differences between group A (oxygen-ozone only) and group C (oxygen-ozone + corticosteroids) in the mean NRS score were found only at the first follow-up visit 30 days post-treatment. This indicates that the corticosteroid component in group C has an additional early-onset effect, thereby reducing LBP earlier than in group A, which has been treated with oxygen-ozone only. However, it is essential to understand that statistical significance observed at the 30-day mark does not necessarily translate to clinical importance. Treating LBP requires a tailored approach, considering each patient’s desires and comfort levels. For instance, while some patients prioritize minimal injections, choosing to forgo the additional corticosteroid injection, others may opt for the best possible pain alleviation, even if short-lived. The early-onset and short-lasting therapeutic effect due to the anti-inflammatory effects of corticosteroids is reflected in the fact that the lowest average scores of NRS and GPS in group B were reported 30 days after the treatment. These results indicate that the effects of corticosteroids are early-onset and short-lasting, while the effects of oxygen-ozone tend to be late-onset but long-lasting. Therefore, both oxygen-ozone and corticosteroids have a synergistic effect for the treatment of LBP. This is consistent with the previous literature suggesting this combination for effective treatment of LBP [29,42]. Gallucci et al. explored this intricate relationship in a study involving 159 patients. In their research, patients were divided into two groups and subjected to different treatments: one group received corticosteroids alone, while the other group received a combination of corticosteroids and oxygen-ozone therapy. Immediate short-term results between both treatment groups were similar. After three months, the clinical effect began to differ but was not statistically significant, while after six months, 74% of the treatment group with corticosteroids + oxygen-ozone had an Oswestry Disability Index of less than 20%, while only 47% of the group with corticosteroids reached this level of recovery [29].

The lasting benefits of oxygen-ozone therapy, along with its broad spectrum of therapeutic applications, can be comprehended by recognizing its diverse properties. These properties encompass local and systemic antimicrobial effects, antioxidant capabilities, anti-inflammatory properties, improvements in microcirculation and localized oxygen supply, reduction in ischemia and nerve edema, muscle pain alleviation and relaxation, and the reduction of glycosaminoglycans (GAGs), resulting in decreased water retention and the replacement of fibrous tissue [43,44,45]. These effects are to be considered an advantage when compared to radiofrequency-pulsed or neuromodulation treatment approaches, as the latter two have not been shown to have such biostimulatory effects. The main therapeutic mechanism for intradiscal injection in herniated discs remains the reduction of GAGs with subsequent water loss, disc volume reduction, and, ultimately, fibrous replacement. Therefore, the effect of intradiscal oxygen-ozone is both mechanical (i.e., reduction of disc volume) and anti-inflammatory. As a result, compression on adjacent nerves and the associated demyelination processes are diminished [41,42,46]. It has been proposed that most of oxygen-ozone’s working effects can be made use of in intradiscal injections, leading to the greatest pain reduction in patients with discopathy [47]. The results presented are in line with a previously published systematic review that compared the effects of oxygen-ozone therapy from randomized controlled trials across a total of 2597 patients. The authors emphasized the safety of the treatment and its non-inferiority in pain control and functional recovery at short to medium-term follow-up but also mentioned the poor methodologic quality of previous studies [28].

No complications or adverse events were reported over the follow-up period of the study, which is in line with previous studies underlining the safety of this treatment [25,26]. While generally considered safe with low rates of complications and adverse events, the treating physician should still keep in mind that severe complications after percutaneous application of oxygen-ozone can still occur in rare cases. Among others, such complications can comprise allergic reactions, spinal cord or nerve injury, the formation of prevertebral abscesses, air embolism, and cardiopulmonary arrest with pneumoencephaly [48,49,50,51].

One strength of this study is the long follow-up period of two years, which facilitated the understanding of the long-term effects of oxygen-ozone in comparison to corticosteroids in the treatment of LBP. Despite being less precise when compared to CT imaging, the use of more radiation-sparing imaging (e.g., fluoroscopy, ultrasound) needs to be analyzed in future randomized controlled trials. Additionally, more studies are required to identify the most optimal medication dosage, timeline, and patient selection to further optimize the described combination therapy and to design standardized treatment protocols.

## 5. Limitations

However, this study is not free of limitations. Firstly, the implementation of functionality scores such as the Oswestry Disability Index would have provided the possibility to better understand the effect of oxygen-ozone on the patient’s functionality associated with LBP. Secondly, a thorough morphological analysis of the MRI and CT images, as described in a previous study, would have allowed us to objectify the treatment results [37]. Stratification based on the morphologic evaluation would have allowed the identification of subgroups with better or worse outcomes to ultimately tailor the treatment based on the presenting morphology. Thirdly, the implementation of a local anesthetics group would have been beneficial to compare the study results with another therapy commonly employed for the treatment of LBP. Fourthly, despite the intended injection site being the intervertebral space for intradiscal product application, this objective was not consistently met. While such cases were rare, there was a lack of precise documentation regarding the exact percentage of instances in which the product deviated from the intended administration site. It can be speculated that non-intradiscal product application (paravertebral, intradiscal, intraforaminal, or periradicular) might result in different treatment outcomes and might be better suited for a different patient population. However, future randomized controlled trials will need to expand on the results presented herein to provide reliable results on alternate injection locations. Fifthly, no information was collected on additional medications used by the patients during the study period, which might have potentially influenced the pain levels following the treatment outcome. Lastly, the uneven distribution of study participants among the three groups can be considered a limitation of this study. This discrepancy was influenced by certain patient characteristics relevant to treatment selection, and, as a result, caution is warranted when generalizing our findings to populations with more balanced group sizes. Future studies will need to substantiate the findings of our study in larger study samples with an even distribution of participants among treatment groups.

## 6. Conclusions

Percutaneous application of oxygen-ozone in patients with LBP showed long-lasting significant improvement of clinical results (e.g., pain reduction, improvement of functionality) up to two years after the treatment. The combination of oxygen-ozone and corticosteroids can work synergistically for LBP, as the corticosteroids show an earlier onset of treatment effects while the oxygen-ozone permits a long-lasting improvement after treatment.

## Figures and Tables

**Figure 1 diagnostics-13-03370-f001:**
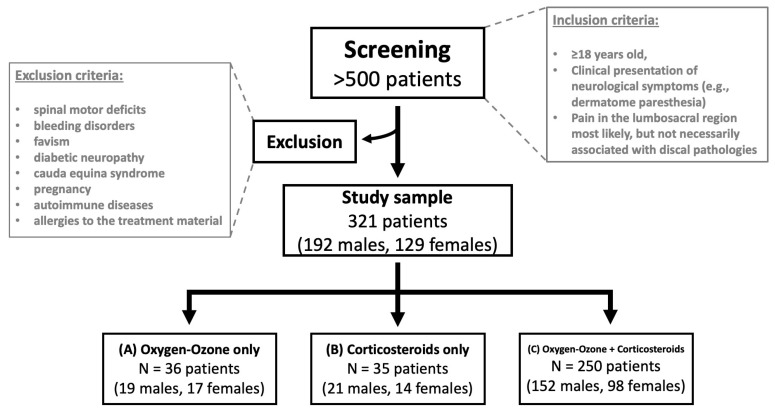
Flowchart summarizing the inclusion and exclusion criteria of the screened study sample and their division into three study arms: (**A**) oxygen-ozone only, (**B**) corticosteroids only, and (**C**) oxygen-ozone + corticosteroids.

**Figure 2 diagnostics-13-03370-f002:**
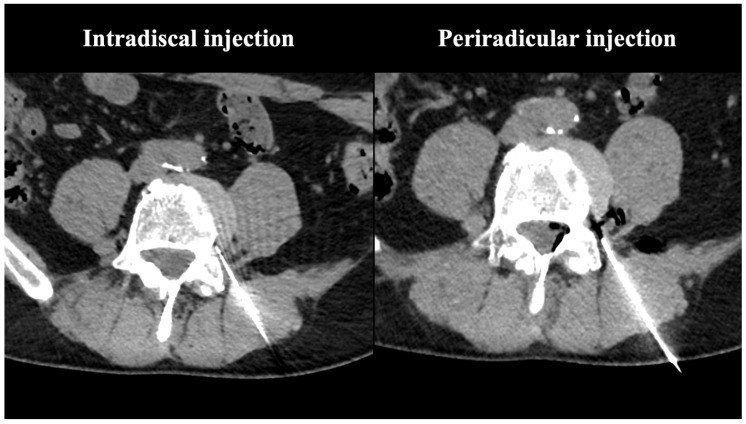
CT scans showing the intradiscal and periradicular injection location of oxygen-ozone.

**Figure 3 diagnostics-13-03370-f003:**
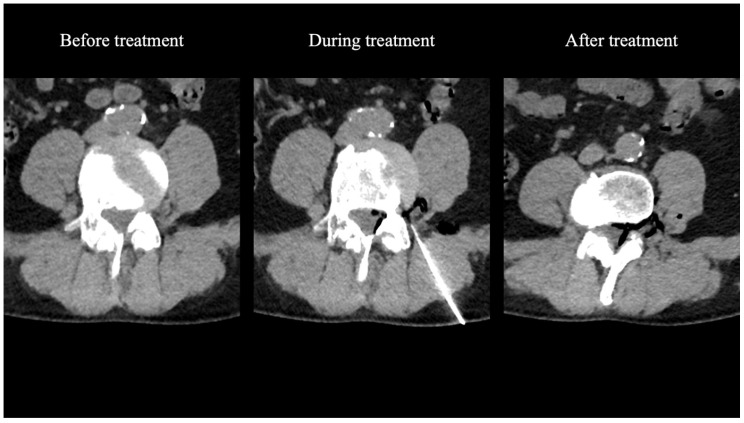
CT scans of the L4/L5 region of a 41-year-old male patient receiving a minimally invasive injection of oxygen-ozone before (left image), during (middle image), and after (right image) the treatment. The correct application of oxygen-ozone is confirmed via the identification of air (i.e., hypodensities) in the middle and right image.

**Figure 4 diagnostics-13-03370-f004:**
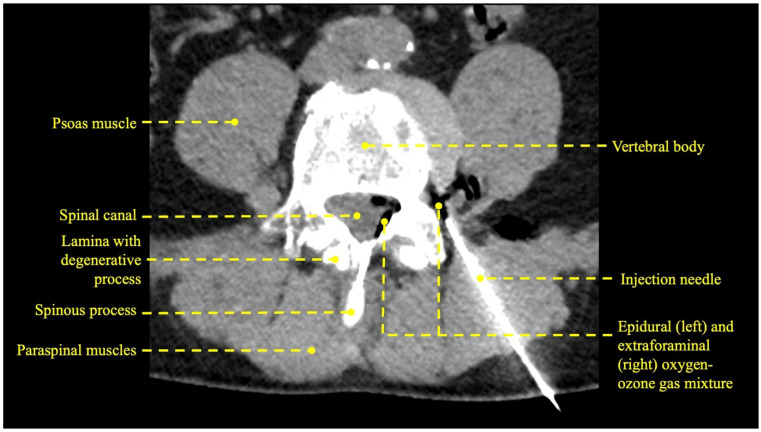
CT scan of the L4/L5 region of a 41-year-old male patient showing the relevant anatomy during the periradicular injection of oxygen-ozone.

**Figure 5 diagnostics-13-03370-f005:**
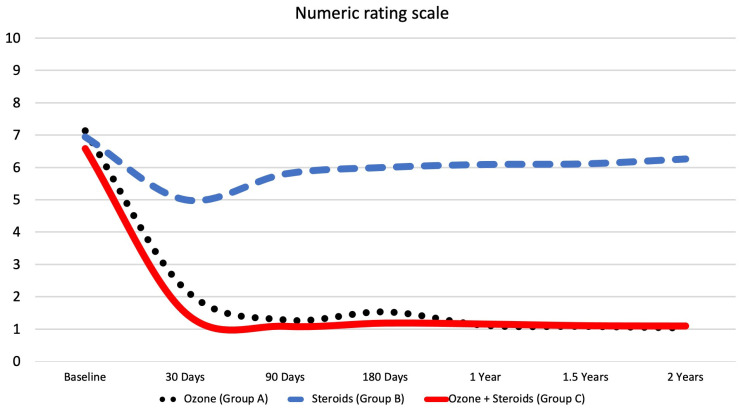
Line graph showing the mean score of the numerical rating scale (NRS) for each study group over the entire follow-up time period.

**Figure 6 diagnostics-13-03370-f006:**
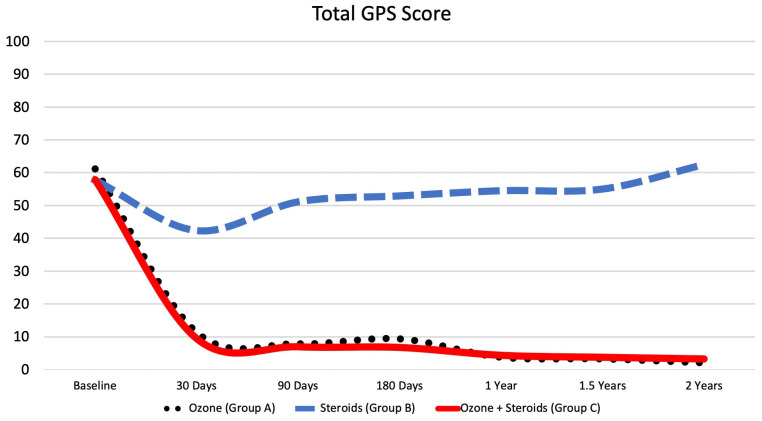
Line graph showing the mean score of the total global pain scale (GPS) for each study group over the entire follow-up time period.

**Figure 7 diagnostics-13-03370-f007:**
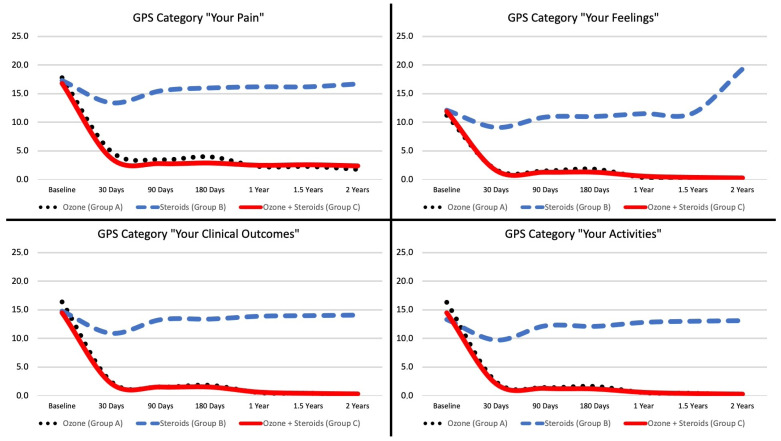
Line graph showing the mean score of each sub-category of the global pain scale (GPS) for each study group over the entire follow-up time period.

**Table 1 diagnostics-13-03370-t001:** Summary of patient data for each of the three study groups: (**A**) oxygen-ozone, (**B**) corticosteroids, (**C**) oxygen-ozone + corticosteroids.

	(A) Oxygen-Ozone Group A *N* = 36	(B) Steroids Group B *N* = 35	(C) Oxygen-Ozone + Steroids Group C *N* = 250
**Gender**			
Male (*N* =192)	19 (52.8%)	21 (60.0%)	152 (60.8%)
Female (*N* = 129)	17 (47.2%)	14 (38.9%)	98 (39.2%)
**Age**			
Age Mean ± SD (years)	52.9 ± 13.3	51.7 ± 18.5	51.3 ± 14.9
**Duration of** **lower back pain**			
Less than 6 months	3 (8.3%)	4 (11.4%)	48 (19.2%)
6–12 months	2 (5.6%)	4 (11.4%)	41 (16.4%)
12–24 months	4 (11.1%)	5 (14.3%)	44 (17.6%)
More than 24 months	27 (75.0%)	22 (62.9%)	117 (46.8%)
**Patient history**			
Previous therapy (Yes)	17 (47.2%)	21 (60.0%)	119 (47.6%)
Previous surgery (Yes)	4 (11.1%)	5 (14.3%)	15 (6.0%)
Previous physical therapy (Yes)	13 (36.1%)	19 (54.3%)	133 (53.2%)

**Table 2 diagnostics-13-03370-t002:** Outcome parameters at baseline, after 30 days, after 90 days, after 180 days, after one year, after one and half years, and after two years for each study group. Bold *p* values indicate a statistically significant difference between the respective follow-up period and baseline.

		Baseline	30 Days	*p* Value	90 Days	*p* Value	180 Days	*p* Value	1 Year	*p* Value	1.5 Years	*p* Value	2 Years	*p* Value
**Oxygen-ozone**	Numeric rating scale	7.1 ± 2.3	2.2 ± 2.3	**<0.001**	1.3 ± 2.1	**<0.001**	1.5 ± 2.3	**<0.001**	1.1 ± 2.3	**<0.001**	1.1 ± 2.2	**<0.001**	1.0 ± 2.2	**<0.001**
Group A	Total GPS Score	61.1 ± 22.7	11.3 ± 11.3	**<0.001**	7.8 ± 10.4	**<0.001**	9.3 ± 12.3	**<0.001**	3.7 ± 6.9	**<0.001**	3.2 ± 6.8	**<0.001**	2.0 ± 4.4	**<0.001**
	“Your Pain”	17.8 ± 5.0	4.9 ± 5.1	**<0.001**	3.5 ± 5.2	**<0.001**	4.0 ± 5.6	**<0.001**	2.3 ± 4.7	**<0.001**	2.3 ± 4.7	**<0.001**	1.8 ± 3.9	**<0.001**
	“Your Feelings”	11.2 ± 6.1	1.7 ± 2.6	**<0.001**	1.5 ± 1.9	**<0.001**	1.8 ± 2.5	**<0.001**	0.3 ± 0.9	**<0.001**	0.1 ± 0.5	**<0.001**	0.0 ± 0.0	**<0.001**
	“Your Clinical Outcomes”	16.4 ± 6.9	2.4 ± 2.9	**<0.001**	1.5 ± 2.4	**<0.001**	1.8 ± 2.9	**<0.001**	0.5 ± 1.0	**<0.001**	0.4 ± 1.1	**<0.001**	0.1 ± 0.3	**<0.001**
	“Your Activities”	16.3 ± 7.8	2.4 ± 2.9	**<0.001**	1.4 ± 2.4	**<0.001**	1.6 ± 2.7	**<0.001**	0.5 ± 1.1	**<0.001**	0.4 ± 1.2	**<0.001**	0.1 ± 0.4	**<0.001**
**Corticosteroids**	Numeric rating scale	6.9 ± 2.4	5.0 ± 2.2	**<0.001**	5.8 ± 1.9	**<0.001**	6.0 ± 2.0	**<0.001**	6.1 ± 2.0	**<0.001**	6.1 ± 1.9	**<0.001**	6.3 ± 1.8	**<0.001**
Group B	Total GPS Score	57.7 ± 21.2	42.3 ± 22.4	**<0.001**	51.1 ± 16.6	**<0.001**	52.9 ± 19.1	**<0.001**	54.5 ± 19.3	**<0.001**	55.0 ± 18.9	**0.002**	62.5 ± 17.0	**<0.001**
	Total GPS Score	17.3 ± 5.0	13.4 ± 5.4	**<0.001**	15.5 ± 3.9	**<0.001**	16.0 ± 4.3	**0.002**	16.2 ± 4.2	**0.004**	16.2 ± 4.1	**0.007**	16.7 ± 4.9	0.328
	“Your Pain”	12.1 ± 7.3	9.1 ± 6.7	**<0.001**	10.9 ± 6.0	**0.013**	11.0 ± 6.3	**0.008**	11.5± 6.5	0.214	11.6 ± 6.5	0.313	19.3 ± 4.2	**<0.001**
	“Your Feelings”	14.8 ± 6.5	10.9 ± 6.6	**<0.001**	13.3 ± 5.7	**<0.001**	13.4 ± 6.2	**<0.001**	13.9 ± 6.2	**<0.001**	14.0 ± 6.1	**<0.001**	14.1 ± 6.3	**<0.001**
	“Your Clinical Outcomes”	13.3 ± 7.0	9.7 ± 6.7	**<0.001**	12.2 ± 6.1	**0.005**	12.1 ± 6.4	**0.018**	12.8 ± 6.6	0.080	13.0 ± 6.4	0.207	13.1 ± 6.5	0.413
**Oxygen-ozone + Corticosteroids**	Numeric rating scale	6.6 ± 2.2	1.5 ± 2.0	**<0.001**	1.1 ± 1.7	**<0.001**	1.2 ± 2.0	**<0.001**	1.2 ± 2.0	**<0.001**	1.1 ± 2.0	**<0.001**	1.1 ± 2.0	**<0.001**
Group C	Total GPS Score	57.9 ± 22.2	9.5 ± 14.5	**<0.001**	7.0 ± 12.1	**<0.001**	6.8 ± 13.6	**<0.001**	4.4 ± 8.8	**<0.001**	3.8 ± 7.6	**<0.001**	3.3 ± 6.5	**<0.001**
	Total GPS Score	16.8 ± 4.8	3.7 ± 5.1	**<0.001**	2.8 ± 4.3	**<0.001**	2.9 ± 4.9	**<0.001**	2.5 ± 4.7	**<0.001**	2.6 ± 4.6	**<0.001**	2.4 ± 4.4	**<0.001**
	“Your Pain”	11.9 ± 6.9	1.6 ± 3.4	**<0.001**	1.3 ± 3.0	**<0.001**	1.3 ± 3.4	**<0.001**	0.6 ± 1.7	**<0.001**	0.4 ± 1.5	**<0.001**	0.3 ± 1.3	**<0.001**
	“Your Feelings”	14.5 ± 6.8	2.1 ± 3.4	**<0.001**	1.5 ± 2.8	**<0.001**	1.5 ± 3.2	**<0.001**	0.6 ± 1.7	**<0.001**	0.4 ± 1.4	**<0.001**	0.3 ± 1.0	**<0.001**
	“Your Clinical Outcomes”	14.5 ± 7.0	2.1 ± 3.8	**<0.001**	1.3 ± 2.8	**<0.001**	1.2 ± 3.1	**<0.001**	0.6 ± 1.5	**<0.001**	0.4 ± 1.1	**<0.001**	0.3 ± 1.0	**<0.001**

## Data Availability

The study data are available from the corresponding author upon reasonable request.

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
