# Peer review of "Percutaneous Computed Tomography-Guided Oxygen-Ozone (O2O3) Injection Therapy in Patients with Lower Back Pain—An Interventional Two-Year Follow-Up Study of 321 Patients"

_diagnostics, 2023, doi:10.3390/diagnostics13213370_

Round 1
Reviewer 1 Report
Comments and Suggestions for Authors
Thank you for sending this interesting research article for our review.
My only concern is significant difference in number of participants in Groups A, B and C with the group C being about 9-10 times larger than other groups. I believe this would be a significant confounding factor, affecting the entire results reliability.
I am wonder if there is any solution to address this issue and make the number of cases in each group more even?
Author Response
Reviewer 1
Reviewer comment
Thank you for sending this interesting research article for our review.
My only concern is significant difference in number of participants in Groups A, B and C with the group C being about 9-10 times larger than other groups. I believe this would be a significant confounding factor, affecting the entire results reliability.
I am wonder if there is any solution to address this issue and make the number of cases in each group more even?
Author response
The authors would like to thank the reviewer for the favorable comment.
We acknowledge your concern regarding the discrepancy in group sizes, and we understand that it may raise questions about the reliability of our results. However, we would like to provide some context regarding the study design and the constraints we faced:
At the time of the study, the allocation of participants into the respective groups was done according to the specific criteria outlined in the methods section and was not influenced by the current research question or hypothesis. Given the ethical considerations and logistical challenges involved, we are unable to alter the group sizes at this stage. Changing group sizes would require a new study with fresh participant recruitment, which may not be feasible or ethical, especially if it involves interventions or invasive procedures.
Despite the uneven group sizes, we have employed rigorous statistical methods to analyze the data and control for potential confounding factors. We believe that these statistical methods have helped mitigate the impact of the initial group size differences on the reliability of our results.
Author action
Following the reviewer’s comment, we have included this limitation in the discussion section.
Please find changes on page 16, line 23 to page 17, line 3:
“Lastly, the uneven distribution of study participants among the three groups can be considered a limitation of this study. This discrepancy was influenced by certain patient characteristics relevant to treatment selection. As a result, caution is warranted when generalizing our findings to populations with more balanced group sizes. Future studies will need to substantiate the findings of our study in larger study samples with even distribution of participants among treatment groups.”
Reviewer 2 Report
Comments and Suggestions for Authors
Dear Authors,
I congratulate you on this very long study.
However, there are some aspects that require your attention.
First of all you need to format the paper using the MDPI standards.
At the end of the article you need to introduce the sections regarding the authors contribution and ethical statements.
Also in the results sections in the Figures 5, 6 , 7 you have a poor choice of a color for illustrating the Ozone Group A, instead of yellow you should use black or another stronger color.
Moreover, explain if the subgroup receiving steroids underwent other procedures, because the follow up is for 2 years, it is very difficult for a patient to continue suffering from pain and not receive another procedure.
In the discussion section underline the possible complications of this procedure such as possible allergic reactions. Reference this to the work by Berghi et al. https://www.researchgate.net/publication/361114470_Current_Approach_to_Medico-Legal_Aspects_of_Allergic_Reactions
Furthermore format the Reference according to MDPI instructions.
At the end of the discussions insert a subheading about the limitations of the present study.
Please clarify the statement at the end of the article: the products utilized in this study were donated by the injectors for the purposes of this study. Were the patients receiving the same substances for injection from the same producer or you used different substances from different procedures? This should also be clarified in the materials and method section.
Looking forward to receiving the improved version of your manuscript.
Comments on the Quality of English LanguageNeeds English Language review as none of the authors is a native speaker of English. Some of the words are correct but their meaning is out of context such as the word affords used in the Conclusion, actually the correct term is permits.
Author Response
Reviewer 2
Reviewer comment
Dear Authors,
I congratulate you on this very long study.
However, there are some aspects that require your attention.
First of all you need to format the paper using the MDPI standards.
At the end of the article you need to introduce the sections regarding the authors contribution and ethical statements.
Author response
The authors would like to thank the reviewer for the time to review our manuscript and for the comment. The reviewer is correct that the current formatting does not align with the MDPI standards.
Author action
According to the reviewer’s comment, the entire manuscript has been re-formatted to now align with the MDPI standards. We also included the author’s contribution, funding statement, ethical statement, conflicts of interest and a data availability statement at the end of the article.
Please find changes throughout the entire manuscript.
Reviewer comment
Also in the results sections in the Figures 5, 6 , 7 you have a poor choice of a color for illustrating the Ozone Group A, instead of yellow you should use black or another stronger color.
Author response
The reviewer is absolutely correct. Yellow has been a poor choice of colour for these figures. For the revision, the authors chose black as a colour to indicate the Oxygen-Ozone group.
Author action
Following the reviewer’s valid comment, we have changed the colour for the Oxygen-Ozone group (Group A) in the figures from yellow to black.
Please find the changes in Figures 5, 6 and 7.
Reviewer comment
Moreover, explain if the subgroup receiving steroids underwent other procedures, because the follow up is for 2 years, it is very difficult for a patient to continue suffering from pain and not receive another procedure.
Author response
The authors would like to use this opportunity to thank the reviewer for the attentive reading. Detailed feedback like yours is essential to improve the quality of our manuscript.
To clarify, the subgroup in question comprises patients who specifically opted against any other form of treatment. These patients occasionally took painkiller medication (e.g., NSAIDS) but consistently declined any other form of treatment, as noted in their informed consent. Consequently, this is the reason for the smaller size of this subgroup. Furthermore, they did not undergo any surgical procedures during the study period.
Author action
To adress the reviewer’s comment, the authors decided to overall clarify the inclusion of patients into the three study arms. For this, we rewrote the methods section.
Please find changes on page 7, line 18-20:
“Patients were treated based on a combination of their individual preferences and the medical indications deemed suitable by the treating physician, ensuring an individualized and patient-centric approach to care while aligning with established therapeutic guidelines.”
Reviewer comment
In the discussion section underline the possible complications of this procedure such as possible allergic reactions. Reference this to the work by Berghi et al. https://www.researchgate.net/publication/361114470_Current_Approach_to_Medico-Legal_Aspects_of_Allergic_Reactions
Author response
We appreciate the reviewer's insightful comment. While our initial search did not yield specific literature on allergic reactions associated with oxygen-ozone injections, we have broadened our discussion on the complications linked to oxygen-ozone treatments. Furthermore, we have incorporated the possibility of oxygen-ozone triggering allergic reactions, and this is now referenced in the updated manuscript.
Author action
Following the reviewer’s suggestion, we have focused more on the complications inherent to oxygen-ozone injection therapy in the discussion section.
Please find changes on page 15, line 19-21:
“Among others, such complications can comprise allergic reaction, spinal cord or nerve injury, the formation of prevertebral abscesses, air embolism, and cardiopulmonary arrest with pneumoencephaly. (49–52)”
Reviewer comment
Furthermore format the Reference according to MDPI instructions.
Author response and action
Once again we would like to thank the reviewer for helping us to improve the quality of our manuscript for potential publication in MDPI. Following the reviewer’s suggestion, we have re-formatted the entire manuscript to align with the MDPI instruction. Please find the changes highlighted throughout the entire manuscript.
Reviewer comment
At the end of the discussions insert a subheading about the limitations of the present study.
Author response and action
The authors have incorporated the suggested change into the most current version of the manuscript. Thank for your attentive reading.
Reviewer comment
Please clarify the statement at the end of the article: the products utilized in this study were donated by the injectors for the purposes of this study. Were the patients receiving the same substances for injection from the same producer or you used different substances from different procedures? This should also be clarified in the materials and method section.
Looking forward to receiving the improved version of your manuscript.
Author response and action
The reviewer is correct that this statement was misleading and did not accurately depict the study's reality. We have improved that statement by incorporating the names and companies of the two products used for our study in the materials and methods section.
Please find the information on page 7, line 21-24:
“The oxygen-ozone gas mixture was produced using an ozone generator OZO2 (Alnitec S.R.L., Cremosano, Italy), and the concentration of the oxygen-ozone gas mixture was monitored throughout the process. The corticosteroids used in this study were Dexason 4mg/mL (Galenika, Belgrade, Serbia).”
Reviewer 3 Report
Comments and Suggestions for Authors
This follow-up study determined the effectiveness of two-year oxygen-ozone use and steroid injection treatment in low back pain patients.
The contents of the review are as follows.
- In Table 2, the results of post-analysis should also be expressed along with the p-value. It is difficult to find a significant difference between Group A and C from the contents of Figure alone.
- As a result of this study, Group A and C only showed differences in numerical rating scale and total GPS for 30 days. Statistically, the difference between the two may be meaningful, but from a clinical point of view, is it meaningful to endure the pain of a steroid injection and receive treatment for this minor difference?
I would like to hear the views of researchers on this. In addition, it is necessary to mention this in the discussion.
Author Response
Reviewer 3
Reviewer comment
This follow-up study determined the effectiveness of two-year oxygen-ozone use and steroid injection treatment in low back pain patients.
The contents of the review are as follows.
- In Table 2, the results of post-analysis should also be expressed along with the p-value. It is difficult to find a significant difference between Group A and C from the contents of Figure alone.
Author response
The reviewer is absolutely correct. We would like to thank for the attentive reading and for the opportunity to substantially improve the quality of our work. We could not agree more that the inclusion of p values is essential. In fact, only one parameter has been shown to be statistically significant different between group A and group C: NRS score at 30 days. This has been described in the manuscript.
Author action
Following the reviewer’s suggestion, we have added p values (i.e., Group A vs Group B, Group B vs Group C and Group A vs Group C) in the revised version of Table 2.
Reviewer comment
As a result of this study, Group A and C only showed differences in numerical rating scale and total GPS for 30 days. Statistically, the difference between the two may be meaningful, but from a clinical point of view, is it meaningful to endure the pain of a steroid injection and receive treatment for this minor difference?
I would like to hear the views of researchers on this. In addition, it is necessary to mention this in the discussion.
Author response
The authors want to thank the reviewer for the valuable feedback on our manuscript. An important question was raised regarding the clinical significance of the observed differences between Group A (oxygen-ozone only) and Group C (oxygen-ozone + corticosteroids) in terms of the numerical rating scale (NRS) scores at the 30-day follow-up.
While we acknowledge the statistical significance of the NRS score difference, we recognize that statistical significance does not always equate to clinical significance. In our study, the choice between corticosteroid injections and oxygen-ozone therapy should consider various factors beyond the 30-day NRS score difference. Our study primarily aimed to explore the long-term effects of these treatments.
For this, the following key points need to be considered:
Statistically significant differences were observed in the 30-day NRS scores, favoring corticosteroid injections (Group C). However, the statistical significance of this difference may not justify choosing corticosteroids over oxygen-ozone therapy alone.
Our study highlights that corticosteroids have an early-onset effect, providing quicker pain relief, while oxygen-ozone therapy offers long-lasting benefits. The combination of both therapies showed a synergistic effect.
Treatment decisions should be individualized, considering both short-term and long-term benefits, patient-specific factors, and the need for immediate pain relief.
We agree that this aspect needs to be expanded on in the revised version of the manuscript.
Author action
Following the reviewer’s suggestion, we have expanded on how statistical significance does not equate clinical significance and how treatment for LBP should be individualized according to the patient’s preferences and wishes.
Please find changes on page 13, line 24 – page 14, line 5:
“However, it is essential to understand that statistical significance observed at the 30-day mark does not necessarily translate to clinical importance. Treating LBP requires a tailored approach, considering each patient's desires and comfort levels. For instance, while some patients prioritize minimal injections, choosing to forgo the additional corticosteroid injection, others may opt for the best possible pain alleviation, even if short-lived.“
Round 2
Reviewer 1 Report
Comments and Suggestions for Authors
Thank you for providing us with the revision of your manuscript. Although the novelty of this manuscript is suboptimal, I believe your study has the potential to add to the literature.
Good Luck.
Reviewer 3 Report
Comments and Suggestions for Authors
Everything has been properly revised.